# Evaluation of a Brief Sodium Screener in Two Samples

**DOI:** 10.3390/nu11010166

**Published:** 2019-01-14

**Authors:** Christy C. Tangney, Heather E. Rasmussen, Candace Richards, Michelle Li, Bradley M. Appelhans

**Affiliations:** 1Department of Clinical Nutrition, Rush University Medical Center, Chicago, IL 60612, USA; heather.rasmussen@unl.edu (H.E.R.); candace.leigh20@gmail.com (C.R.); Michelle_Li@rush.edu (M.L.); 2Department of Nutrition & Health Sciences, University of Nebraska–Lincoln, Lincoln, NE 68588, USA; 3Department of Nutrition Services, Aurora St. Luke’s Medical Center, Milwaukee, WI 53215, USA; 4Department of Preventive Medicine, Rush University Medical Center, Chicago, IL 60612, USA; brad_appelhans@rush.edu

**Keywords:** sodium screener, agreement, 24-h recall, sodium guidelines

## Abstract

The Sodium Screener© (SS©), as developed by NutritionQuest (Berkeley, CA, USA), was designed to reduce the burden of repeated dietary or urinary sodium measurements, but the accuracy of daily sodium intake estimates has not been reported. Associations were examined between sodium intakes derived from the SS© scores and repeated 24-h recalls (24DR) in two studies with different administration modes. In one study, 102 registered dietitians (RD) completed three Automated Self-Administered 24DRs (ASA24©), version 2014, followed by the SS©; both were self-administered and web-based. In the second sample, (the Study of Household Purchasing Patterns, Eating, and Recreation or SHoPPER), trained dietitians conducted 24DR interviews with 69 community-dwelling adults in their homes; all the community adults then completed a paper-based SS© at the final visit. In the RD study, SS© -predicted sodium intakes were 2604 ± 990 (mean ± Standard deviation (SD)), and ASA24© sodium intakes were 3193 ± 907 mg/day. In the SHoPPER sample, corresponding values were 3338 ± 1310 mg/day and 2939 ± 1231 mg/day, respectively. SS©-predicted and recall sodium estimates were correlated in the RD study (*r* = 0.381, *p* = 0.0001) and in the SHoPPER (*r* = 0.430, *p* = 0.0002). Agreement between the SS© and 24-h recalls was poor when classifying individuals as meeting the dietary sodium guidelines of 2300 mg/day or not (RD study: kappa = 0.080, *p* = 0.32; SHoPPER: kappa = 0.207, *p* = 0.08). Based on repeated 24DR either in person or self-reported online as the criterion for estimating daily sodium intakes, the SS© may require additional modifications.

## 1. Introduction

Considerable efforts are being made to reduce sodium consumption at both the individual and the population level, and practical brief tools to monitor sodium intakes are needed. Urinary sodium excretion is considered by many to be a “gold” standard for assessing the sodium intake of population samples, but it is time-consuming, costly, and not without measurement error [1]. At present, most dietary sodium assessments, such as 24-h dietary recall, require time for administration, completion, and subsequent diet analyses. There is a need for dietary sodium assessment tools that provide rapid feedback to clients in order to foster changes in specific dietary behaviors. A sodium screener that can be scored by the health professional and, possibly, by the respondent would be useful in clinical settings; however, considerable bias in sodium intake estimates may arise from the need to include a limited number of food items on the screener and use “generic” item descriptions (i.e., with specific brands excluded). 

The objective of this research was to assess the accuracy of the brief Sodium Screener© (SS©) developed by NutritionQuest (Berkeley, CA, USA) against a criterion method of repeated 24-h recalls. We compared these instruments in two contexts. In the first, the sample was composed solely of registered dietitians (RD) who completed both the SS© and repeated 24-h recalls through web-based software. In the second, adults in the community completed the SS© on paper, after repeated 24-h diet recall interviews were conducted by a RD. The two methods allowed for an evaluation of the SS© in a highly standardized context involving expert respondents, and for a setting that emulated the context in which health professionals discuss dietary sodium intake with their clients by using expert dietary interviewers. 

## 2. Materials and Methods

*Participants.* The first sample was comprised of 102 RDs from the Sodium Screener Validation Study. No dietary intervention was used in this study; the aim was to test how the screener performed in parallel with more traditional dietary assessment methods. These assessments were conducted exclusively online from September 2014 to March 2015. A systematic random sample of practicing RDs in the US with daily internet access was acquired from a list by the Commission on Dietetic Registration. We will refer to this as the **RD** study. All RDs were sent an information sheet approved by the Institutional Review Board and a link to a Research Electronic Data Capture or REDCap (version 6.3.0, Nashville, TN, USA) demographic survey. Completion of the survey was considered consent to participate. 

The second sample was drawn from the Study of Household Purchasing Patterns, Eating, and Recreation (SHoPPER; ClinicalTrials.gov identifier: NCT02073643 [2]). The sample included primary household shoppers living in Chicago who collected household food and beverage purchase receipts. They were interviewed by RDs in their homes four times over two weeks. Participants completed several questionnaires during these visits. During the latter part of subject accrual for SHoPPER, the NutritionQuest Sodium Screener© (SS©) was added to the study. A paper copy of the SS© was given to all participants following the third or fourth 24-h recall interview.

*Demographic information.* In the RD study, a REDCap survey was sent to all RD participants to capture key demographic characteristics of respondents including date of birth, dates unavailable to complete questionnaires, sex, ethnicity, race, and others. Similar demographic information (sex, race-ethnicity and date of birth) as well as other information (education, income, household size, etc.) was acquired from SHoPPER participants using standard questionnaires. 

### 2.1. Brief Sodium Screener© (SS©)

*Background on the Sodium Screener (SS©).* The 26-item SS© was developed by Block et al. (see Reference [3,4]) using 24-h recall data from adults in National Health and Nutrition Examination Survey (NHANES) 2007–2008. Foods contributing to 80% of sodium intake were included in the SS©, in which there are five frequency response categories—from rarely or never (assigned a value of 0) to every day (assigned a value of 4). Usual portion size is queried for seven items. For each, the respondent chooses one of three options (with values 0, 1, or 2). The cumulative SS© score ranges from 0 to 67, with higher scores signifying higher sodium intakes. SS© scores are interpreted on the basis of a hand-scoring key provided on the tool. Sodium intake is judged to be less than 2300 mg per day if the scores range between 15 and 22 (males) or 18 and 24 (females). If the scores equal or exceed these cutoffs, sodium intakes are above target and deemed as not meeting the dietary guidelines. The SS© scores may also be used in one of two sex-specific prediction equations to obtain an estimate of daily sodium intake, as described by NutritionQuest. These equations were derived from full-length Block Food Frequency Questionnaire data and adjusted to approximate the median levels of adults in NHANES 2007–2008 [4]. 

*Administration of the SS©.* In the RD study, the SS© was made into an electronic survey instrument within the REDCap project. In the SHoPPER, a paper copy of the identically-worded SS© was given to the participant to complete during the home visit. 

### 2.2. Criterion Methods

*RD Study*. The criterion was multiple (at least three) non-consecutive and web-based Automated Self-Administered 24-h recall or ASA24© (version 2014). ASA24© was developed by the National Cancer Institute under contract with the research firm Westat (Rockville, MD, USA). ASA24© uses the automated multi-pass method and the Food and Nutrient Database of Dietary Studies, version 5, to compute reported food and nutrient intakes [5]. Participants select a “talking” avatar (a penguin) or read the questions themselves on the screen. The two researchers (ML and CR) assigned two weekdays and one weekend day to each participant using the REDCap calendar and automated email invitations. The RD respondents were not aware that the SS© was being evaluated. 

*SHoPPER.* The criterion was multiple in-person 24-h recall interviews conducted by trained dietitians on scheduled home visits. (Three interviews were required; a fourth would occur if daily intake was atypical.) Recalls were acquired using the Nutrition Data System for Research or NDSR (versions 2014–2016, Nutrition Coordinating Center, University of Minnesota, Minneapolis, MN, USA), a web-based software in which the automated multiple-pass method was applied. The dietitians who interviewed the participants were not aware that sodium intakes were of interest to the investigators.

### 2.3. Statistical Analyses

Analyses were conducted using SPSS, version 22 (IBM SPSS, Armonk, NY, USA), separately for each study. The first analyses were those of classification or agreement, because we wanted to know whether the SS© correctly classified participants as meeting or not meeting the 2300 mg dietary sodium guideline. First, we compared the number of participants who met or did not meet the 2300 mg cutoff based on predicted sodium intakes derived from the SS© with estimates from averaged recall sodium intakes. These comparisons replicate an approach likely used by providers in clinics. Sensitivity, specificity, percent agreement, and kappas were calculated for both samples. 

Next, concordance between sodium intakes derived from recalls with those predicted from the SS© was evaluated with Pearson correlation tests; these tests were run with de-attenuation for random error in 24-h recalls. De-attenuation was calculated based on the formula provided by Beaton et al. [6]. Finally, to obtain a visual assessment of the extent of bias between the recall and screener estimates of sodium intakes, Bland–Altman plots, as in Reference [7], were constructed. 

## 3. Results

Of the 1575 RDs contacted, a total of 191 participants responded to the demographic survey (12.1%). A final number of 102 respondents completed all three recalls. In the SHoPPER, all 69 participants completed the SS© after being asked at the third or fourth home visit. 

As shown in Table 1, the participants in both studies were mostly women. The SHoPPER sample was more diverse in terms of race, though similar in age. The distribution of SS© scores was normal in both. Estimates for sodium intakes on the basis of multiple recalls in both samples are higher than recommended (2300 mg/day). On the basis of recall data, 16.7% of RDs met the guidelines by consuming less than 2300 mg sodium per day, while nearly one third of SHoPPER participants reported consuming less than 2300 mg of sodium daily (30.4%). In contrast, based on the SS© responses in the RD sample, 42.2% reported consuming diets containing an average of less than 2300 mg sodium per day, while in the SHoPPER sample, 20.3% met the guideline. 

As shown in Table 2, there was 58.8% agreement (kappa = 0.08, *p* = 0.32) between the two methods in categorizing participants in the RD study as meeting or not meeting the dietary sodium guideline. In the SHoPPER, there was 69.5% agreement (kappa = 0.207, *p* = 0.08) between the two methods in categorizing adults as meeting or not meeting the cutoff of 2300 mg of daily sodium. Correlations between the two tools were low or fair in both samples. De-attenuated coefficients were higher than unadjusted coefficients.

Bland–Altman plots are shown in Figure 1A,B for the RD and SHoPPER samples, respectively. Bias is presented as the difference between the sodium intakes estimated by the SS© using the gender-specific equations provided and the average of three non-consecutive 24-h recalls. The mean bias between the two was −550 ± 1048 mg dietary sodium per day in the RD sample and +399 ± 1309 mg dietary sodium per day in the SHoPPER sample. This shows that, relative to diet recalls, the SS© underestimated sodium intakes reported by RDs using the ASA24© recalls and slightly overestimated sodium intakes for community adults. Limits of the agreement (± 2 SD) were similarly wide across the two study samples: ± 2096 mg/day for the RD study and ± 2620 mg/day in the SHoPPER. 

## 4. Discussion

The objective of this research was to determine whether a brief screener correctly identifies individuals as consuming high sodium intakes in comparison to those identified based on repeated 24-h recalls. We expected the classification based on SS© scores (whether raw score or predicted sodium intakes from those scores) to agree with the time-intensive open-ended 24-h recall method. There was poor to moderate agreement in classifying participants as meeting or not meeting the 2300 mg sodium per day guideline in both studies. Sensitivity and concordance were moderate, but bias was great when sodium intakes were examined as continuous variables. 

As stated previously, the SS© can be scored immediately following completion, thus informing health professionals whether the dietary goal was met and alerting them to certain sodium-rich foods consumed frequently by participants. While a brief screening tool can never be expected to capture all the possible food sources of sodium, it is important to have some confidence in the derived estimates. The time saved on the part of the participant and the health professional cannot be justified if one has little confidence that the screener accurately captures dietary sodium intakes. Thus, one cannot be confident whether the participant has been adherent to sodium guidelines or not. 

In the RD study, we compared daily sodium estimates from a brief screener with one that necessitates the recording of food and beverage amounts online in a sample of dietitians who had at least a college education. Considerable food expertise was needed to perform daily recalls online; this was intentional in order to provide the best-case scenario to test the web-based SS© and ASA24©. In SHoPPER, dietitians conducted recall interviews with adults in the home setting, which is highly desirable. Both approaches could provide insights into the dietary behaviors of two very different samples in relation to sodium intakes. However, our criterion method of repeated 24-h recall is not without bias, as shown by several researchers [8,9]. 

The observed bias and limits of agreements as shown in Figure 1A,B were most striking. For the RD group, SS©-predicted values underestimate sodium intake compared to the 24-h recall estimates. It is unclear why this underestimation occurred. One possibility is that food items on the SS© did not capture the main sources of sodium for the RDs. RD respondents may have provided more detail and, therefore, a more complete estimate of what they ate (especially with regards to sodium content) in their web-based recalls than the community adults (on which the SS© scoring equations are based). Based on SS© scores, more RDs were deemed accordant (42%) compared to accordance based on averaged recalls (17%). For the SHoPPER group, a smaller proportion of adults was deemed accordant based on the SS© (20%) when compared to the averaged recall (30%). 

We anticipated concordance between predicted sodium intake from the SS© or the SS© scores and intakes from the averaged recall to be higher. There are several possible reasons for the observed discrepancy between these methods. Perhaps the non-specific food items on the SS© do not reflect the greatest contributors to overall sodium intake in the two samples examined, especially in the RD sample. It is noteworthy that over a decade has passed since the SS© was developed, the main contributors to dietary sodium may have shifted since that time. Additionally, the variability in sodium content across brands of products can be considerable [10]. Possibly, preparation details afforded by the recall method as provided by the RDs, who presumably are more aware of the sodium content of foods, significantly differed from the sodium content of items that appear on the SS©. It is also possible that the differences in the food composition databases (ASA24© and NDSR) used for the 24-h recalls reflect a more up-to-date analysis of sodium content compared with what was used to construct the SS©.

This research has several strengths. The same screener was applied to two samples similar in age but different in ethnicity, gender, presumably different in nutrition knowledge, and possibly different in educational background. Had concordance been stronger and bias smaller, the use of the two different samples would have provided support for the general use of this screener. We recommend that further assessments of the SS© are made, including validation against repeated 24-h urinary sodium assessments.

## Figures and Tables

**Figure 1 nutrients-11-00166-f001:**
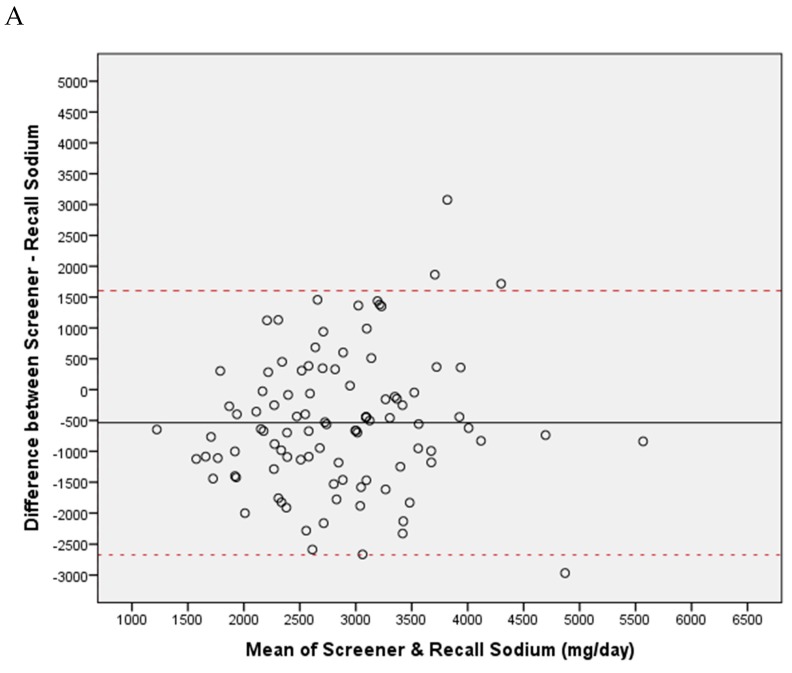
A Bland–Altman graph of bias (mean difference) between the predicted sodium estimates derived from the Sodium Screener (SS©) and dietary sodium intake from the average of three recalls in the RD study (**A**) and in the SHoPPER (**B**). The dashed lines represent limits of agreement (± 2 SD). Predicted intakes from the SS© scores underestimated daily sodium intake by 550 mg (**A**) and overestimated daily sodium intake by 399 mg (**B**). Bias did not appear to vary by the amount consumed.

**Table 1 nutrients-11-00166-t001:** Demographic characteristics and dietary estimates based on the Sodium Screener© and averaged 24-h recalls in two different study samples.

**Characteristics**	**Registered Dietitian (RD) Screener Comparison Study** **(*n* = 102)**	**SHoPPER Community Sample** **(*n* = 69)**
Women (%)	97.1	85.5
Race (%)		
White	89.2 ^1^	37.7 ^1^
Black	3.9	36.2
Asian	2.9	4.3
Hispanic	2.0	18.8
Age (year)	43 ± 14 ^2^	45 ± 14
% with College Degree	100	53.5
**Sodium Screener (SS©) Information**		
Use Low Sodium Foods *Often* or *Always* (%)	59.6	46.3
*Never* Use Table Salt (%)	56.9	36.2
Sodium Screener Score (SS©)	27 ± 9 11–54 ^3^	34 ± 11 4–61 ^3^
Predicted Sodium Intake from SS© (mg)	2604 ± 990 ^2^	3338 ± 1231
**Average Intake based on analysis of three dietary recalls**	**ASA24-2014 ^4^ 24-h recalls (self-administered)**	**NDSR ^4^ 24-h recalls by interviewers**
Energy (kcal)	1821 ± 481	1935 ± 609
Sodium (mg) ^5^	3193 ± 907	2939 ± 1223
Sodium mg/1000 kcal	1773 ± 328	1513 ± 384

^1^ Additional race categories included for the registered dieticians (RD) study were native American (1.2%); multi-ethnic (2.9%); and, for the SHoPPER, native American (2.9%), multi-ethnic (10.17%), and unable to choose (8.8%). ^2^ Three ages were not provided; calculations were for only 99 participants. ^3^ Range (minimum-maximum). ^4^ ASA24©: Automated Self-Administered 24-h Recall Software, version 2014, from the National Cancer Institute; NDSR: Nutrition Data System for Research (Nutrition Coordinating Center, University of Minnesota). ^5^ Reported dietary sodium estimates from the average of three recalls were unadjusted.

**Table 2 nutrients-11-00166-t002:** Agreement, sensitivity, specificity, and concordance between methods used to estimate dietary sodium intakes.

Characteristics	Registered Dietitian (RD) Screener Study (*n* = 99)	SHoPPER Community Sample (*n* = 69)
Agreement (A) between daily sodium intakes predicted by the SS© ^1^ and those from the average of three recalls (number meeting/not meeting the 2300 mg cutoff) ^2^ Kappa		

	58.8	69.5

	0.080 (0.32) ^3^	0.207 (0.08)
Sensitivity/Specificity (%/%)	60/53	85/67
Correlations: predicted sodium from the SS© with averaged sodium intake based on three recalls Crude De-attenuated not energy adjusted		

	0.381 (0.0001) ^3^	0.430 (0.0002)
	0.633	0.849

^1^ SS©: Sodium Screener©. ^2^ Agreement between actual SS© scores with recall average was similar. For the RD study, there was 57.8% agreement, kappa = 0.072 (0.371), and crude correlation, *r* = 0.337 (0.001); for the SHoPPER study, there was 68.1% agreement, kappa = 0.181 (0.122), and crude correlation *r* = 0.342 (0.004). ^3^
*p*-value is in parentheses.

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
