# Peer review of "Evaluation of a Brief Sodium Screener in Two Samples"

_nutrients, 2019, doi:10.3390/nu11010166_

Reviewer 1 Report

This study described the accuracy of the SS against repeated 24-hour recall for the assessment of daily sodium intake. The purpose and methods are clear and reasonable. Unfortunately, bias: the difference between sodium intakes estimated by SS and 24-hour recall was not small, sensitivity and concordance of high sodium intake (> 2300 mg) was moderate. I agree with the conclusions of the authors that the SS may merit additional modifications, but I accept the results as they are.

Author Response

There was no request for changes. Thank you for your comments!

Reviewer 2 Report

General Comment:  The study compared estimates of sodium intake via the Sodium Screener (SS) method with two criterion methods (RD & SHoPPER) in two separate samples of people (registered dietitians for RD study and adults for the SHoPPER study). Results are well presented and consistent with regard to the main finding that the SS does not provide a high level of confidence in estimating sodium intake.  The paper is concise, clear, and well written.  I agree with the author’s interpretations presented in the paragraph in lines 207-218 that suggest there might be some outdated issues related to the SS.  If SS data have not been previously analyzed as the authors have presented, this information is quite important for health care professionals. 

I would ask the authors to consider the following points. 

1.    Are the authors comfortable with making a stronger conclusion?  Statistically, the study clearly shows that the limits of agreement are wide and bias is high.  Clinically, the questions become how wide and how high?  Based on the kappa values, there is slight, or at best, only fair agreement between SS and RD & SHoPPER results, respectfully.  I agree that further assessments of SS are warranted as the authors state (line 222).  Do the statistics justify recommending discontinuing the SS as a method for estimating sodium intake?  If not, I would suggest saying that based on the results of this study, health professionals utilizing SS to estimate sodium intake must use caution before recommending dietary or life-style changes. 

2.    It appears there is a trend in both of the Bland-Altman graphs showing that the difference between methods tend to get larger as the average increases.  This strengthens the argument that the two methods are not equivalent. 

3.    The authors do state that participants from both studies, especially the RD study, are mostly women.  A comment on why this is occurred might be helpful.  Were the percentages of women participants in the Sodium Screener Validation Study and the SHoPPER trial the same as appeared in Table 1?  Any reason to suspect that the same study with men, who probably consume more sodium, would yield different results?

Author Response

1.        Are the authors comfortable with making a stronger conclusion?

The reviewer has asked whether we should state our concerns more strongly. We are concerned that this tool be carefully considered if it is to be used to guide subsequent dietary intervention or counseling.  While the correlations between the sodium screener estimates and recall estimates are reasonable (r’s=~0.3-0.4), kappas are poor.  What we don’t know yet is whether the tool is responsive to dietary changes in sodium intakes, especially if such changes are captured by change in the frequency of specific food items on the screener. With such a wide limits of agreement in both samples, such changes might not be reflected in change in screener scores.  We prefer to have others evaluate the tool to ascertain whether such observations are universal. (NO changes have been made to the Manuscript in response to this point)

2.       It appears there is a trend in both of the Bland-Altman graphs showing that the difference between methods tend to get larger as the average increases. This strengthens the argument that the two methods are not equivalent.

The bias appears to definitely increase with greater intakes in Figure 1B. (No changes to manuscript are indicated in response to this comment)

3.       The authors do state that participants from both studies, especially the RD study, are mostly women. A comment on why this is occurred might be helpful. Were the percentages of women participants in the Sodium Screener Validation Study and the SHoPPER trial the same as appeared in Table 1?

Practicing RDs in the US acquired from lists provided by the Commission on Dietetic Registration are overwhelmingly female. Since the sample was a systematic random sample, no effort was made to recruit separately by sex.  Although the community sample in SHoPPER had more men proportionally, the eligibility for the sample was contingent on being the predominant person in charge of food purchases for the household. The percentages are as stated in Table 1.

4.

Any reason to suspect that the same study with men, who probably consume more sodium, would yield different results?

    If the study was repeated in a sample with a higher proportion of men, we are certain if the kappa’s would improve or not. We believe there are too few men to adequately examine the kappa’s, correlations, and Bland Altman graph separately in SHoPPER. Nor did we measure knowledge of sodium content in food among men and women in our study, so it is not possible to conjecture whether different outcomes would be possible.